# Changes in dietary habit and physical activity among the diabetic patients of Bangladesh during COVID-19: A cross-sectional study

Ishrat Jahan[1], A. B. M. Nahid Hasan[1], Azaz Bin Sharif[1,2]*, Sharmin Akter[3]

**1** Department of Public Health, North South University, Dhaka, Bangladesh, **2** Global Health Institute, North South University, Dhaka, Bangladesh, **3** Nutrition and Health Education Department, Bangladesh Institute of Health Science General Hospital, Dhaka, Bangladesh

☯ These authors contributed equally to this work.
* azaz.sharif@northsouth.edu

**Data Availability Statement:** The dataset is attached as a supporting document.

**Funding:** The authors received no specific funding for this work.

## Abstract

In an effort to avert further Covid-19 transmission, the Bangladesh government took several initiatives which disrupted the routine food intake and exercise of diabetic patients. This study sought to examine the difference in dietary and exercise habits of diabetic patients between their pre-pandemic status and during COVID-19 which may be attributed to the observed poor health outcomes during the study period. This was a cross-sectional study that enrolled 604 diabetic patients using a convenience sampling technique who were attending outpatient clinics of the three selected hospitals in Bangladesh. A validated semi-structured questionnaire was used to collect information regarding eating habit and physical activity of the respondents pre- and during COVID-19 pandemic via direct interview. McNe-mar-Bowker test was used to assess the changes in dietary and physical activity behaviors. The current study reveals that 93.9% of the respondents were type-2 diabetic patients. During the pandemic, there was a decrease in the consumption of rice, bread, meat, fish, eggs, and dessert, while the consumption of cereals, milk, and potato/starchy vegetables increased. There was also a decrease in the frequency of drinking tea or coffee, while the consumption of soft drinks remained relatively stable. The level and duration of physical activity significantly decreased among the respondents during the pandemic. This study explored the changes in dietary habits and physical activity among the study population which not only disrupt the metabolic control of the diabetic patients but also pose a significant threat to their overall health. Therefore, it is critical to prioritize measures that support diabetic patients to maintain healthy dietary habit and to engage in regular physical activity during unprecedented times such as COVID-19 pandemic.

## Introduction

Coronavirus disease 2019 (COVID- 19) was identified and disclosed for the first time in Wuhan, China, at the end of 2019, which was caused by "Severe Acute Respiratory Syndrome

**Competing interests:** The authors have declared that no competing interests exist.

Coronavirus 2 (SARS CoV-2) viral infection, and rapidly spread around the globe [1]. In response to the rapid spread of the disease, governments around the world resolved to implement emergency measures such as lockdowns, isolation, quarantine, and social and physical distancing to interrupt community transmission of the virus [2]. On March 26, 2020, the Bangladesh government issued an executive order declaring a state of emergency, following the implementation of extreme lockdown measures such as social isolation and house confinement to limit the spread of the virus and prevent the health system from crumbling [3]. While these restrictions are meant to prevent infection, they may have the unintended consequence of limiting routine daily activities, physical activity (PA), travel, and the availability of a diverse variety of dietary options [4].

A balanced, healthy diet is a major element of a personal risk management plan during pandemics, such as the COVID-19 pandemic [5]. The World Health Organization (WHO) has recognized the importance of diet and physical activity in the prevention and treatment of non-communicable chronic diseases [6]. Many organizations have issued food and nutrition suggestions during the lockdown, as there is a strong link between diet quality and overall health [7], especially in type 2 diabetes mellitus patients. A Polish online survey among diabetic patients in 2020 concluded that, only about 60% of the respondents had consumed more regular and nutritious diet during the pandemic [8]. During confinement, food accessibility could be hampered, which might have a negative impact on total diet quality [9].

Lockdowns drastically decreased physical activity, sport, and exercise, resulting in negative health impacts, such as sedentarism [10]. An observational study among diabetic adults in Netherland revealed that, 45.7% of the respondents had decreased their level of physical activity during the pandemic [11]. Another study among 72 Spanish diabetic people observed that, a significant proportion of the respondents avoided physical activity during COVID-19 [12].

COVID-19 had also put a negative effect on increasing sedentary behavior up to 28% [9]. A recent meta-analysis suggested that sedentary behavior, such as, prolonged television-viewing time was associated with an increased risk of type 2 diabetes, cardiovascular disease, and all-cause mortality. Maintaining such behaviors even for a short period of time (a few weeks) has been shown to have negative metabolic consequences (such as poor glycemic control, increased body fat, increased abdominal fat, and increased inflammatory cytokines) that have an effect on the management of diabetes and other NCDs. Moreover, it is imperative to emphasize the importance of patient education and self-care promotion for effective management of their disease [13].

Patients with COVID-19 who also have uncontrolled diabetes, face a significantly increased chance of dying. A recent study found thatCOVID-19 patients with diabetes had a higher risk of death, a greater number of organ injuries, and a greater need for medical treatments [14]. The result of an global online survey indicated that during COVID-19 home confinement respondents significantly decreased their levels of physical activity, increased their daily sitting time, and increased their consumption of unhealthy food and ate less regularly [9]. Exercise and balanced diet are essential for the management of diabetes. Disruptions in recommended dietary habit, and physical activity might have serious health consequences as well as worsening diabetes' metabolic control [15].

A significant portion of Bangladesh's population struggles to attain financial stability [16]. The spread of COVID-19 has impacted not only the health of the people in Bangladesh, but also the country's economy and food supply. Therefore, the poor suffer greatly due to their lack of resources and healthcare. People all over the world have been instructed to stay indoors due to the COVID-19 lockdown, which has made it impossible to go outside. Most people in lower middle-income countries (LMICs) like Bangladesh, consisting of a huge populations, cannot afford spacious places because of their low incomes [17]. In general, due to economic

inflation during COVID-19, most diabetic patients in these countries were unable to leave their homes for exercise or walking, and they also could not afford readily available food.

In light of the aforementioned discussion, it is essential to evaluate the impact of the COVID-19 pandemic on the diabetes mellitus (DM) patients' dietary habit, as well as, physical activity levels. Therefore, the aim of this study was to investigate the changes in dietary habits and physical activity of diabetic patients in Bangladesh during COVID-19 pandemic.

## Materials and methods

### Study setting and population

A cross-sectional study was conducted from January 2022 to June 2022 in the outpatient departments (OPD) of three hospitals (Bangladesh Institute of Health Science Hospital, Dhaka, Thakurgaon Diabetic Hospital, and Jamalpur Medical College Hospital) of Bangladesh that treats diabetic patients. Based on location, patient volume, and diabetes expertise, the authors picked these three hospitals. This study included 604 patients who were suffering from Type 1 (T1) or Type 2 (T2) diabetes for at least one year. A convenient sampling technique was used to select the respondents whose age were between 28 to 72 years. Time, resources, and other constraints limited the author's ability to obtain a random sample. Patients with diabetes who were eligible and available during OPD hours were referred to data collectors by OPD physicians and invited to participate in the study. Respondents who were living overseas, pregnant women, those with COVID symptoms, psychiatric patients, and mentally retarded people were excluded. The sample size was calculated using the formula of Cochran's $\left(n = \frac{z^2 * p(1-p)}{e^2}\right)$ With 5% of margin of error (e), estimated population proportion 26% (p) [18] and standard normal deviation of 1.96 (z), the estimated sample size was 296. The study team tried to reach a large sample size to mitigate possible biases.

### Data collection process

A semi-structured questionnaire was developed in the Kobo Toolbox and the data were collected using a face-to-face interview. The questionnaire was developed based on previously published work, with slight adjustments to the study cohort [9, 19–23]. A nutritionist was involved throughout the development of the questionnaire. The questionnaire was then translated into Bengali, the respondents' native language, and translated back to English. Before collecting data, a pilot test was conducted with 5% of the intended sample size to identify and eliminate any errors or inconsistencies in the questionnaire and to validate the questionnaire.

The questionnaire consisted of three sections; the first section contained questions regarding the participant's socio-demographic status and behavioral questions, second section had diabetes related questions, and the final section contained questions regarding dietary habits, physical activity, and other sedentary activities. Questions related to the changed status of any relevant variables were collected cross-sectionally. In addition to the study's objectives, methods, and anticipated benefits, all respondents were assured of their complete anonymity.

### Variables and measurements

**Primary outcome variable.** Primary outcome variables constitute self-reported dietary habits and self-reported physical activity. Changes in eating habits before or during the pandemic, was measured based on the frequency of consumption of the following products: rice, bread, cereals, pulse and legumes, meat and fish, egg and milk products, vegetables, sweet fruits, dessert, nuts and seeds, refined sugar, tea and coffee, soft drinks, and fast food. Food frequency consumption for these products—with the following responses: "Never", "1–3 times a

**Table 1. Demographic characteristics of the respondents (n = 604).**

| Variable | Categories | n (%) |
|---|---|---|
| **Age(years)** | 28–42 | 100 (16.6) |
| | 43–57 | 273 (45.2) |
| | 58–72 | 231 (38.2) |
| **Gender** | Male | 251 (41.6) |
| | Female | 353 (58.4) |
| **Religion** | Muslim | 584 (96.7) |
| | Hindu | 19 (3.1) |
| | Others | 1 (0.2) |
| **Education level** | Illiterate | 107 (17.7) |
| | Primary | 167 (27.6) |
| | SSC | 90 (14.9) |
| | HSC | 70 (11.6) |
| | Graduate and above | 170 (28.1) |
| **Occupation** | Government service holder | 51 (8.4) |
| | Private service holder | 93 (15.4) |
| | Business | 71 (11.8) |
| | Housewife | 291 (48.2) |
| | Others (Jobless, retired, farmers, students) | 98 (16.2) |
| **BMI (kg/m$^2$)** | Underweight | 29 (4.8) |
| | Normal | 293 (48.5) |
| | Overweight | 214 (35.4) |
| | Obese | 68 (11.3) |
| **Change of body weight during the pandemic** | No | 244 (40.4) |
| | Yes, it has increased < = 5 kg | 162 (26.8) |
| | Yes, it has increased > 5 kg | 39 (6.5) |
| | Yes, it has decreased < = 5 kg | 92 (15.2) |
| | Yes, it has decreased > = 5 kg | 67 (11.1) |
| **Smokers** | Before the pandemic | 67 (11.1) |
| | During the pandemic | 27 (4.5) |
| **Household monthly income (BDT) [Before the Pandemic]** | < = 10,000 | 146 (24.2) |
| | 10,001–40,000 | 198 (32.7) |
| | 40,001–70,000 | 178 (29.5) |
| | 70,001–1,00,000 | 55 (9.1) |
| | >1,00,000 | 27 (4.5) |

(*Continued*)

**Table 1.** (Continued)

| Variable | Categories | n (%) |
|---|---|---|
| **Household monthly income (BDT) [During the Pandemic]** | < = 10,000 | 217 (36.0) |
| | 10,001–40,000 | 214 (35.4) |
| | 40,001–70,000 | 127 (21.0) |
| | 70,001–1,00,000 | 35 (5.8) |
| | >1,00,000 | 11 (1.8) |

Note: SSC: Secondary School Certificate; HSC: Higher Secondary School Certificate; BMI: Body Mass Index; BDT: Bangladeshi Taka

month", "1–3 times a week", "4–6 times a week", "Once a day", "Few times a day." The type of physical activities included walking, running and others (Dancing, treadmill, swimming, gymnastics jogging, cycling etc.). The level of physical activity included following responses: "I don't exercise", "<3 days per week", "3–5 days per week", "6 and more days per week." Duration of physical activity included following responses: "I don't exercise", "<30 minutes/day", "30–60 minutes/day", "More than 60 minutes/day."

Socio-demographic variables, included in this study were age, gender, religion, education, occupation, household income, smoking status before and during the outbreak, height, weight, and weight change during the pandemic. Digital scale was used to measure weight and height of the respondents. This study adopted the BMI definition of the World Health Organization (2000). The following World Health Organization-recommended norms were used for adults: underweight (below 18.5 kg/m$^2$), normal (18.5–24.9 kg/m$^2$), overweight (25.0–29.9 kg/m$^2$) and obese (above 30.0 kg/m$^2$). Other disease related explanatory variables were types of diabetes, duration of DM, types of treatment received for DM, and family history of DM. The last segment included sedentary activities related variables, such as duration of sleep, and duration of screen time.

## Statistical analysis

The sociodemographic characteristics and diabetes related variables were reported as frequency distributions, while continuous variables, such as age, weight(kg), height (M) and BMI are expressed as mean and standard deviation. McNemar-Bowker test was used to assess the changes in dietary habits and physical activity pre- and during COVID-19 pandemic period. All tests were assumed as two-tailed and the p-value $< 0.05$ was considered to be statistically significant. Graphical presentations were done for types of PA and duration of screen time using multiple bar graphs. Statistical software SPSS (Version 25) was used for all statistical analyses.

## Ethics statement

The Institutional Review Board (IRB) of North South University (NSU) approved the study (2022/OR-NSU/IRB/0807) with the second author as the principal investigator, and all respondents gave consent to participate. The research complied with the ethical standards established by the three institutions from which the respondents were recruited.

## Results

### Demographic characteristics

The demographic characteristics of the study population is presented in **Table 1**. Majority of the surveyed individuals were aged between 43–57 years (45.2%), female (58.4%), housewives (48.2%), Muslim (96.7%), had a normal BMI (48.5%), and had completed graduation (28.1%). Compared to the pre-pandemic, during pandemic 40.4% of the respondents' body weight had no change, 26.8% of the respondents' body weight increased by 0–5 kg, and 15.2% of the respondents' body weight decreased by 0–5 kg. About 11.1% of the respondents used to smoke before the pandemic which reduced to 4.5% during the pandemic. Majority of the respondent's monthly income was between 10,001–40,000 BDT before the pandemic which has shifted to the income group ≤10,000 BDT (36%).

**Table 2. Distribution of diabetes related factors among the respondents (n = 604).**

| Variable | Categories | n (%) |
|---|---|---|
| **Type of diabetes** | Type-1 | 37 (6.1) |
| | Type-2 | 567 (93.9) |
| **Duration of diabetes** | Under 2 years | 82 (13.6) |
| | 2–5 years | 129 (21.4) |
| | 6–10 years | 157 (26.0) |
| | Over 10 years | 236 (39.1) |
| **Type of treatment** | Insulin pen | 171 (28.3) |
| | Oral anti-diabetic drug | 306 (50.7) |
| | Insulin and oral antidiabetic drug | 127 (21.0) |
| **Family history** | No family history | 291 (48.2) |
| | Had family history | 313 (51.8) |
| **Measurement of blood glucose level within last 3 months** | Yes | 439 (72.7) |
| | No | 165 (27.3) |
| **Frequency of self-monitoring blood glucose level** | Never | 167 (27.6) |
| | Sometimes | 423 (70.0) |
| | < 2 times/day | 10 (1.7) |
| | > = 2 times/day | 4 (0.7) |
| **Disease management during the pandemic** | No | 254 (42.1) |
| | Yes, my disease self-management has improved | 198 (32.8) |
| | Yes, my disease self-management has declined | 152 (25.1) |

## Diabetes related factors

**Table 2** presents information on factors related to diabetes. About 94% of the respondents had type-2 diabetes whereas only 6.1% had type-1 diabetes. In regard to the duration of diabetes, a majority of the respondents had been suffering from diabetes over 10 years (39.1%). Proportion of the respondents had been suffering from diabetes for 6–10 years, 2–5 years, and under 2 years were 26.0%, 21.4%, and 13.6%, respectively. Majority of the respondents (50.7%) were under oral antidiabetic drug, whereas 28.3% had been taking insulin pen, and the remaining respondents (21%) had been taking both. Among the respondents, 51.8% had family history of diabetes. The proportion of the respondents who had measured their blood glucose level within last 3 months was 72.7%. Occasional self-blood glucose level monitoring was reported by 70.0% of the respondents. On the contrary, 27.6% never monitored their blood glucose level. About 25.1% individuals declared that their diabetes was not in-control, 32.8% stated that it had improved, while 42.1% did not report any changes.

**Table 3** presents the frequency of different food items consumption among the study respondents before and during the pandemic. During the pandemic, 8.3% of the respondents had been consuming cereals more than once a week (p < 0.001). Meat and fish intake decreased with 48.2% of respondents consuming them once a day before the pandemic, compared to 40.7% during the pandemic (p < 0.001). Among the respondents, egg consumption during pandemic for at least once a day decreased to 30.3% from pre-pandemic 35.1% (p = 0.001). Milk was consumed by 29.0% of respondents 1–3 times per week before the pandemic, while it was increased to 30.0% during the pandemic (p = 0.011). Before the pandemic, 11.9% of the respondents used to consume potato and starchy vegetables 4–6 times/week which increased to 14.1% during the pandemic whereas their daily intake decreased during the pandemic (p = 0.019). Green and colored vegetables were consumed by 38.4% of respondents few times a day before pandemic, which dropped to 36.6% during pandemic (p = 0.004). Consumption frequency of dessert decreased during the pandemic, with 11.9% of respondents' consuming them 1–3 times per week compared to 15.7% pre-pandemic (p < 0.001). Among the respondents, 33.6% reported drinking tea or coffee once a day before the pandemic, compared to 22% during the pandemic (p = 0.015). The consumption of nuts and seeds remained relatively stable, with 23.2% of respondents consuming them 1–3 times per week before the pandemic, compared to 23.5% during the pandemic (p = 0.012).

## Physical activity

**Fig 1** shows the type of physical activity chosen before and during the COVID-19 pandemic. The bar graph depicts that, 20.2% of the respondents' reported not engaging in any physical activity before the pandemic and the percentage increased to 43.1% during the pandemic. Before the pandemic, 75.7% of respondents engaged in walking as a form of physical activity, but this proportion reduced to 55.1% during the pandemic. 2.5% of the respondents who used to engage in running before the pandemic was decreased to 0.3% during the pandemic.

**Table 4** presents the changes in level and duration of physical activity among the study respondents from pre-pandemic to during pandemic. The percentage of individuals who exercised 3–5 days per week decreased from 20.9% to 18.5%, and those who exercised 6 or more days per week decreased from 52.6% to 28.2% (p < 0.001). The pandemic had an effect on the duration of physical activity as well. The percentage of individuals who exercise 30–60 minutes per day decreased from 39.4% to 23.7%, and those who exercised more than 60 minutes per day dropped sharply from 8.9% to 2.8% (p < 0.001).

**Table 3. Comparison of study respondents ' dietary habits before and during the pandemic (n = 604).**

| Food items | Frequency of food intake | | | | | | p-value |
|---|---|---|---|---|---|---|---|
| | Never | 1–3 times/month | 1–3 times/week | 4–6 times/week | Once a day | Few times/day | |
| *Rice* | | | | | | | |
| Before the pandemic | 0 (0.0) | 0 (0.0) | 6 (1.0) | 1 (0.2) | 306 (50.7) | 291 (48.2) | **0.010** |
| During the pandemic | 0 (0.0) | 0 (0.0) | 8 (1.3) | 13 (2.2) | 296 (49.0) | 287 (47.5) | |
| *Bread* | | | | | | | |
| Before the pandemic | 39 (6.5) | 0 (0.0) | 20 (3.3) | 11 (1.8) | 295 (48.8) | 239 (39.6) | 0.083 |
| During the pandemic | 45 (7.5) | 0 (0.0) | 19 (3.1) | 14 (2.3) | 285 (47.2) | 241 (39.9) | |
| *Cereals* | | | | | | | |
| Before the pandemic | 440 (72.8) | 49 (8.1) | 63 (10.4) | 23 (3.8) | 23 (3.8) | 6 (1.0) | **<0.001** |
| During the pandemic | 456 (75.5) | 34 (5.6) | 50 (8.3) | 35 (5.8) | 22 (3.6) | 7 (1.2) | |
| *Pulse and legumes* | | | | | | | |
| Before the pandemic | 71 (11.8) | 32 (5.3) | 230 (38.1) | 47 (7.8) | 158 (26.2) | 66 (10.9) | 0.148 |
| During the pandemic | 76 (12.6) | 32 (5.3) | 230 (38.1) | 52 (8.6) | 151 (25.0) | 63 (10.4) | |
| *Meat and fish* | | | | | | | |
| Before the pandemic | 0 (0.0) | 24 (4.0) | 132 (21.9) | 91 (15.1) | 291 (48.2) | 66 (10.8) | **<0.001** |
| During the pandemic | 0 (0.0) | 40 (6.6) | 122 (20.2) | 137 (22.7) | 246 (40.7) | 59 (9.8) | |
| *Egg* | | | | | | | |
| Before the pandemic | 38 (6.3) | 41 (6.8) | 243 (40.2) | 64 (10.6) | 212 (35.1) | 6 (1.0) | **0.001** |
| During the pandemic | 40 (6.6) | 48 (7.9) | 245 (40.6) | 79 (13.1) | 183 (30.3) | 9 (1.5) | |
| *Milk* | | | | | | | |
| Before the pandemic | 79 (13.1) | 100 (16.6) | 175 (29.0) | 45 (7.5) | 205 (33.9) | 0 (0.0) | **0.011** |
| During the pandemic | 94 (15.6) | 94 (15.6) | 181 (30.0) | 46 (7.6) | 189 (31.3) | 0 (0.0) | |
| *Potato and starchy vegetables* | | | | | | | |
| Before the pandemic | 68 (11.3) | 58 (9.6) | 187 (31.0) | 72 (11.9) | 183 (30.3) | 36 (6.0) | **0.019** |
| During the pandemic | 72 (11.9) | 59 (9.8) | 185 (30.6) | 85 (14.1) | 168 (27.8) | 35 (5.8) | |
| *Green and colored vegetables* | | | | | | | |
| Before the pandemic | 0 (0.0) | 12 (2.0) | 50 (8.3) | 133 (22.0) | 177 (29.3) | 232 (38.4) | **0.004** |
| During the pandemic | 0 (0.0) | 12 (2.0) | 58 (9.6) | 133 (22.0) | 180 (29.8) | 221 (36.6) | |
| *Sweet fruits* | | | | | | | |
| Before the pandemic | 141 (23.3) | 178 (29.5) | 188 (31.1) | 35 (5.8) | 54 (8.9) | 8 (1.3) | 0.223 |
| During the pandemic | 150 (24.8) | 183 (30.3) | 179 (29.6) | 34 (5.6) | 52 (8.6) | 6 (1.0) | |
| *Dessert* | | | | | | | |
| Before the pandemic | 295 (48.8) | 197 (32.6) | 95 (15.7) | 17 (2.8) | 0 (0.0) | 0 (0.0) | **<0.001** |
| During the pandemic | 311 (51.5) | 207 (34.3) | 72 (11.9) | 14 (2.3) | 0 (0.0) | 0(0.0) | |
| *Nuts and seeds* | | | | | | | |
| Before the pandemic | 193 (32.0) | 151 (25.0) | 140 (23.2) | 32 (5.3) | 83 (13.7) | 5 (0.08) | **0.012** |
| During the pandemic | 202 (33.4) | 134 (22.2) | 142 (23.5) | 43 (7.1) | 78 (12.9) | 5 (0.08) | |
| *Refined sugar* | | | | | | | |
| Before the pandemic | 450 (74.5) | 59 (9.8) | 30 (5.0) | 6 (1.0) | 52 (8.6) | 7 (1.2) | 0.619 |
| During the pandemic | 454 (75.2) | 59 (9.8) | 28 (4.6) | 4 (0.7) | 52 (8.6) | 7 (1.2) | |
| *Tea or coffee* | | | | | | | |
| Before the pandemic | 121 (20.0) | 27 (4.5) | 62 (10.3) | 20 (3.3) | 203 (33.6) | 171 (28.3) | **0.015** |
| During the pandemic | 133 (22.0) | 23 (3.8) | 61 (10.1) | 34 (5.6) | 183 (30.3) | 170 (28.2) | |
| *Soft drinks* | | | | | | | |
| Before the pandemic | 397 (65.7) | 144 (23.8) | 53 (8.8) | 10 (1.7) | 0 (0.0) | 0 (0.0) | 0.331 |
| During the pandemic | 404 (66.9) | 143 (23.7) | 49 (8.1) | 8 (1.3) | 0 (0.0) | 0 (0.0) | |
| *Fast food* | | | | | | | |

(*Continued*)

**Table 3.** (Continued)

| Food items | Frequency of food intake | | | | | | p-value |
|---|---|---|---|---|---|---|---|
| | **Never** | **1–3 times/month** | **1–3 times/week** | **4–6 times/week** | **Once a day** | **Few times/day** | |
| Before the pandemic | 377 (62.4) | 154 (25.5) | 61 (10.1) | 7 (1.2) | 4 (0.7) | 1 (0.2) | **0.005** |
| During the pandemic | 394 (65.2) | 148 (24.5) | 52 (8.6) | 4 (0.7) | 4 (0.7) | 2 (0.3) | |

Data is expressed in frequency and percentages.

McNemar-Bowker test was performed to determine the significant group differences before and during the pandemic.

Significant differences are marked in bold (p < 0.05).

## Screen time

**Fig 2** depicts the percentages of respondents' duration of screen time for work and non-work purposes before and during pandemic. The results show that the proportion of people whose duration of screen time for work for 2–4 hours went up from 3.3% to 4.8%. Also, the number of people who used screens for 5–7 hours and 8 or more hours grew from 0.7% and 0.5% to 2.2% and 1.2%, respectively.

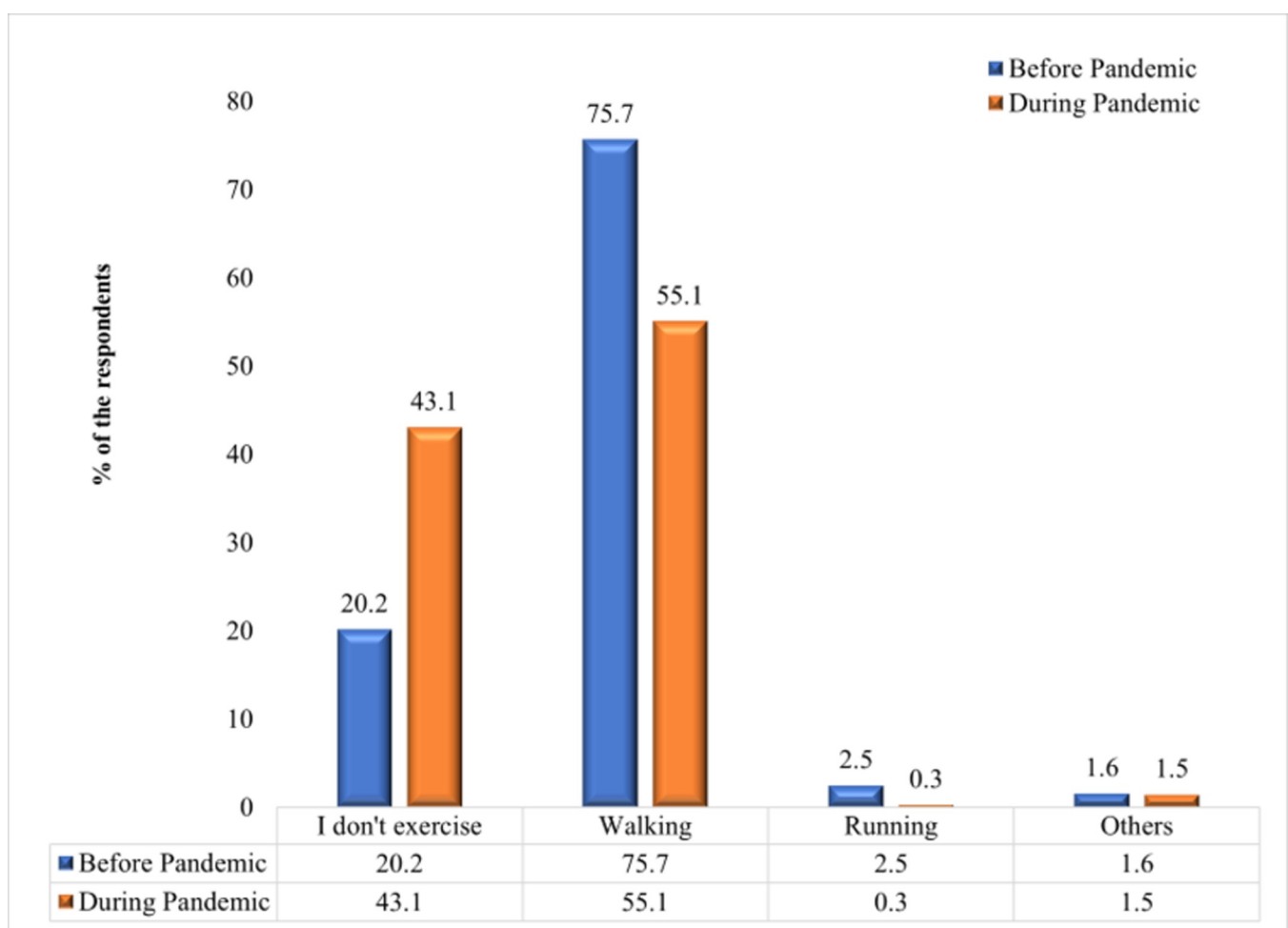

| | I don't exercise | Walking | Running | Others |
|---|---|---|---|---|
| ■ Before Pandemic | 20.2 | 75.7 | 2.5 | 1.6 |
| ■ During Pandemic | 43.1 | 55.1 | 0.3 | 1.5 |

**Fig 1. Types of physical activity before and during the pandemic.**

**Table 4. Changes in level and duration of physical activity of study respondents before and during the pandemic (n = 604).**

**Level of physical activity**

| | Don't exercise | < 3 days/week | 3–5 days/week | 6 or more days/week | p-value |
|---|---|---|---|---|---|
| **Before the pandemic** | 125 (20.7) | 35 (5.8) | 126 (20.9) | 318 (52.6) | **0.000** |
| **During the pandemic** | 260 (43.0) | 62 (10.3) | 112 (18.5) | 170 (28.1) | |

**Duration of physical activity**

| | Don't exercise | < 30 mins/day | 30–60 mins/day | >60 mins/day | p-value |
|---|---|---|---|---|---|
| **Before the pandemic** | 124 (20.5) | 188 (31.1) | 238 (39.4) | 54 (8.9) | **0.000** |
| **During the pandemic** | 260 (43.0) | 184 (30.5) | 143 (23.7) | 17 (2.8) | |

Data is expressed in frequency and percentages.

McNemar-Bowker test was performed to determine the significant group differences before and during the pandemic.

Significant differences are marked in bold (p < 0.05).

## Discussion

To the best of the authors' knowledge, this study is the first in Bangladesh which has looked upon the changes in nutritional and lifestyle aspects of patients with DM before and during the pandemic. The present study demonstrated that there are significant changes in dietary habits and physical activity of diabetic patients during COVID-19 pandemic.

This study revealed that a major proportion of respondents' monthly household income decreased during the pandemic, probably due to the lockdown and economic instability around the world as well as throughout the country. A number of respondents in this study quit smoking during the pandemic. This phenomenon could be explained by the fear induced in smokers of the increased risk of respiratory distress and mortality from COVID-19 [24]. In the current study, most of the respondents had no change of body weight. Similar finding was observed in a South Indian study [25].

In this study, proportion of most of the food intakes decreased during the pandemic among the study respondents compared to the pre-pandemic state. This may be a result of the respondents' economic downturn, which prevented them from maintaining their regular dietary habits. A contradictory result was found in a survey conducted among diabetic patients in Poland which revealed that, over 60% of all respondents started eating more nutritious and regular meals during the COVID-19 pandemic due to their health consciousness during such crisis period [8]. According to the findings of the current study, daily protein consumption such as fish and meat intake as well as milk and egg intake significantly decreased which could be a great threat for the diabetic patients. The study findings also suggest that fresh green and other vegetable consumption had decreased among the diabetic patients and this could be due to enormous difficulties of the agricultural supply chain or difficulties in finding grocery stores open in their neighborhood during the COVID-19, corroborating the aforementioned findings of Bilal et al. [26]. Similar to the results of this study, a decrease in the consumption of fast food/snacks, desserts, sweet fruits and sweet beverages, was observed in an Italian survey [19].

An adverse effect of the COVID-19 pandemic and related government-imposed regulations on mobility was limited outdoor activity. The study found that the percentage of people who did not engage in any form of physical activity has increased during the COVID-19 compared to pre-pandemic period. The findings also suggest that, not only the type of PA but also duration and level of PA significantly decreased among the respondents during this pandemic. Despite given advice that home confinement or lockdown should not withhold people from being physically active [19], present results show that there has been a reduction in all PA levels during the COVID-19. People may be discouraged from engaging in the prescribed amounts

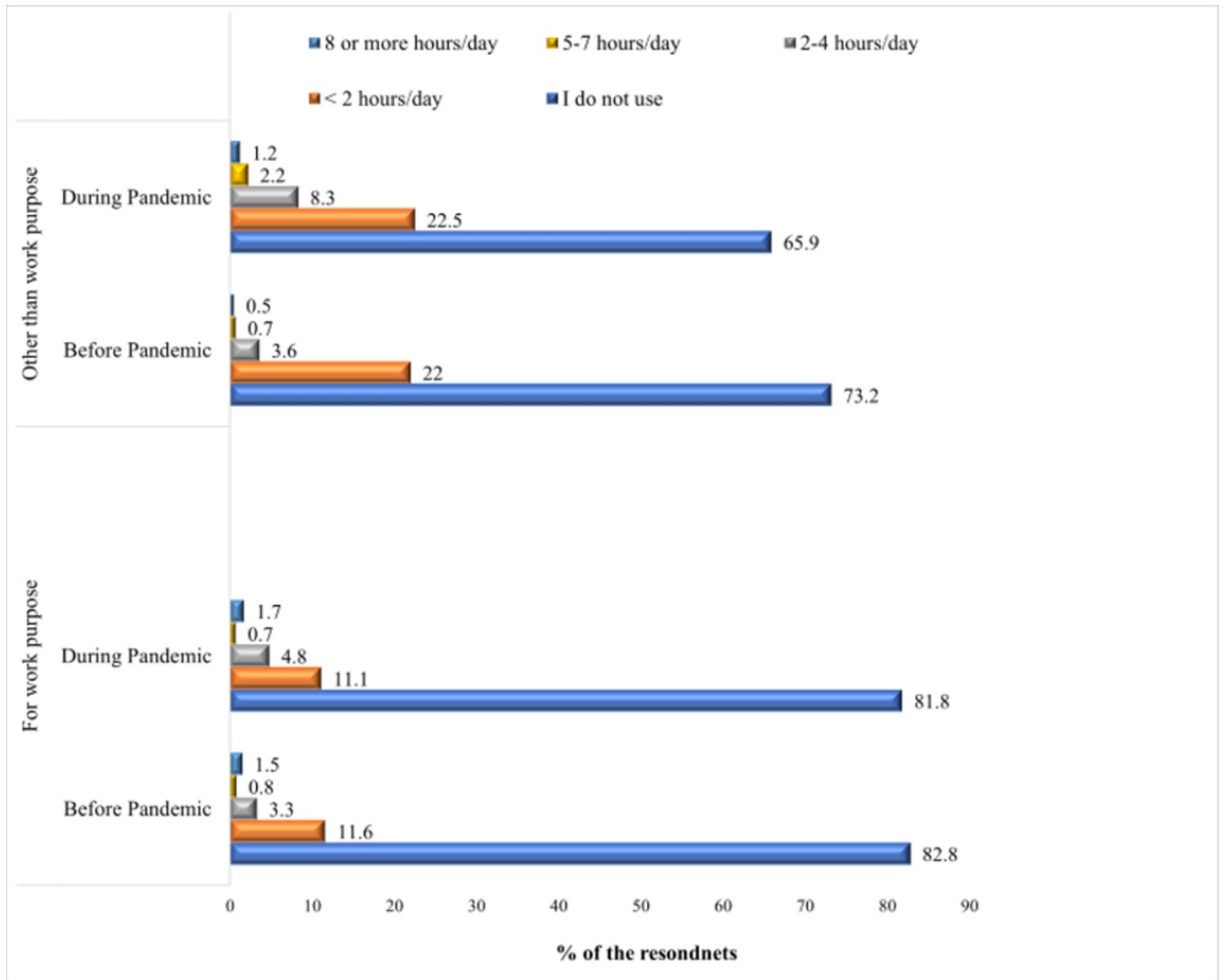

**Fig 2. Duration of screen time before and during the pandemic.**

of physical activity due to the fear of contracting the disease and the mobility limitations enforced during the COVID-19 epidemic. A study conducted in Bangladesh found that nearly 38% of respondents were physically inactive during the COVID-19 [27]. Another Italian study also identified a decrease in the frequency of physical activity during the lockdown [19].

This study also intended to observe the screen time duration during this pandemic, regardless of whether or not it was for work-related purposes. The findings suggest that, while working for 8 or more hours, duration of screen time slightly increased (1.5% vs. 1.7%) during the COVID-19. Additionally, it was noted that respondents' non-work-related screen usage increased. A study among Polish diabetic patients revealed that, the percentage of spending time in front of screen increased to 19% during the pandemic [8]. This indicates that, the COVID-19 pandemic affected regular lifestyle of the people and the level of sedentary behavior increased among them.

The findings from this study could be utilized for planning and promoting dietary advices for diabetic patients during any pandemic crisis. Motivating people to stand up can be a first step of health promotion against sedentary behavior. Further research should address (i) insight into diabetic populations for the development of interventions to address their needs, (ii) interference of diet and PA behaviors, for improving interventions, and (iii) identification of conditions for successfully maintaining a healthy lifestyle before as well as during such pandemic.

The strengths of the study are that, a direct face to face interview with the respondents was conducted compared to telephonic and online interviews prevalent in previous studies. Authors had considered calculating BMI by measuring height and weight physically rather than relying on self-reported measures. The sample size of this study was more than the estimated size. So, this sample size can be expected to represent the diabetic population. Another strength of this study is that the study was carried out in multiple health facilities. As limitations of the study, it should be noted that patients' calorie intake was not measured. The findings are from a well aware population visiting hospital and may not be applicable to the other diabetic population with poor access to healthcare facilities. There might have chance of difficulty to recall dietary and physical activity habits before the COVID-19. Another limitation is that, convenient sampling method may have resulted in selection bias. Moreover, the cross-sectional study design provided information about the status of the study variables in a particular time point, and hence wouldn't facilitate to establish causal inference. Finally, while the present study provides an overview of dietary habits and PA during COVID-19 Pandemic, its results cannot be interpreted in the context of long-term effects.

## Conclusion

This study explored the changes in dietary habits and physical activity among the study population which not only disrupt the metabolic control of the diabetes patients but also pose a significant threat to their overall health. Therefore, it is critical to prioritize measures that supports diabetic patients to maintain healthy dietary habit and to engage in regular physical activity during unprecedented times such as COVID-19 pandemic.

## Supporting information

**S1 Dataset. Dataset containing responses from 604 respondents on their dietary habits and physical activity before and during the COVID-19 pandemic.**
(XLSX)

## Acknowledgments

We thank our consortium's teammates who provided insight and expertise that greatly assisted the research. We are immensely grateful to our data collectors for providing continuous support throughout the research.

## Author Contributions

**Conceptualization:** Ishrat Jahan, A. B. M. Nahid Hasan, Azaz Bin Sharif.

**Data curation:** Ishrat Jahan, A. B. M. Nahid Hasan, Sharmin Akter.

**Formal analysis:** Ishrat Jahan.

**Investigation:** Ishrat Jahan, A. B. M. Nahid Hasan, Azaz Bin Sharif.

**Methodology:** Ishrat Jahan, A. B. M. Nahid Hasan, Azaz Bin Sharif.

**Project administration:** Azaz Bin Sharif.

**Resources:** Sharmin Akter.

**Software:** Ishrat Jahan.

**Supervision:** Azaz Bin Sharif.

**Validation:** A. B. M. Nahid Hasan, Sharmin Akter.

**Visualization:** Sharmin Akter.

**Writing – original draft:** Ishrat Jahan.

**Writing – review & editing:** A. B. M. Nahid Hasan, Azaz Bin Sharif.

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
