## [Decision Letter · Decision Letter 0]

16 Mar 2023

PGPH-D-23-00054

Changes in Dietary Behavior and Physical Activity among the Diabetic Patients of Bangladesh during COVID-19: A Cross-Sectional Study

Dear Dr. Sharif,,

Thank you for submitting your manuscript to PLOS Global Public Health. After careful consideration, we feel that it has merit but does not fully meet PLOS Global Public Health’s publication criteria as it currently stands. Therefore, we invite you to submit a revised version of the manuscript that addresses the points raised during the review process.

Kindly attend to all the issues raised by the reviewers to give strength to the manuscript.

We look forward to receiving your revised manuscript.

Kind regards,

Nnodimele Onuigbo Atulomah, PhD

Academic Editor

Journal Requirements:

1. Please send a completed 'Competing Interests' statement, including any COIs declared by your co-authors. If you have no competing interests to declare, please state "The authors have declared that no competing interests exist". Otherwise please declare all competing interests beginning with the statement "I have read the journal's policy and the authors of this manuscript have the following competing interests:"

Additional Editor Comments:

The study is appropriate and relevant but there are revisions suggested by the reviewers which I am emphasizing that the authors pay particular attention to.

Abstract: The abstract to this study needs to be revised to communicate adequately what the study sought to achieve and what was found. The following represent revised suggestions in the abstract:

Lines 17-25: “In an effort to avert further Covid-19 transmission, the Bangladesh government took several initiatives which disrupted the routine food intake and exercise of diabetic patients. This study sought to examine the difference in dietary and exercise habits these patients between their pre-pandemic status and during COVID-19 which may be attributed to the observed poor health outcomes during the study period.

Methods: This was a cross-sectional study design that enrolled 604 confirmed patients with type- I and type-2 diabetes Mellitus by convenience sampling method attending outpatient clinics in three hospital in Bangladesh. A validated semi-structured questionnaire was developed, using the Kobo toolbox, to collect information by interviewer-administered technique regarding self-reported eating habit and physical activities of patients before and after COVID-19.”

The results and conclusion needs to align with the revised results. The results suggested other factors are involved in the study but are not mentioned.

Introduction

The introduction require revision to strengthen understanding of the context involved.

Lines 46 - 50 should read; "Coronavirus disease 2019 (COVID- 19) was identified and disclosed for the first time in Wuhan, China, at the end of 2019, which was caused by "Severe Acute Respiratory Syndrome Coronavirus 2 (SARS CoV-2) viral infection, and rapidly spread around the globe [1]. In response to the rapid spread of the disease, governments around the world resolved to implement emergency measures such as lockdowns isolation, quarantine, and social and physical distancing to interrupt community transmission of the virus[2].

Materials and methods: Lines 122 to 130: Understanding the descriptions of the variables in the study would benefit greatly if they are organized as "Primary Outcome variable: Self-Reported Dietary Habits" and "Self-Reported Physical Activity" with descriptions of what constitutes each. And Secondary Outcome of "BMI" if measured see line 132.

Statistical Analysis: The focus of the study should guide the data analysis structure. There appears to be a drifting away from the main objectives for the study (see lines 96-99) "The aim of this study was to investigate the changes in dietary habits and physical activity of diabetic patients in Bangladesh during COVID-19 pandemic". Here the study in lines 147-151 is referring to associations instead of comparing pre-Covid-19 outcomes with during Covid-19 values at the time of the study. I believe that Analysis of Variance of Z-test would have been the most appropriate for comparing pre-pandemic dietary experiences and physical experiences with Covid-19 period, Clarity is required here applying the most appropriate statistical tools for comparison rather than correlation.

Lines 143 - 145 should read; "The sociodemographic characteristics and diabetes related variables were reported as frequency distribution while continuous variables of age, weight(kg), height (M) and BMI are expressed as mean and standard deviation.”

Line 145 should read; "Changes in dietary pattern and physical activity before and during the pandemic were categorized as "increased", "decreased", and "No change" and scored as 2-points, 1-point and zero-point for each participants in the study.”

Results:

The results in the manuscript needs to be organized to facilitate clarity and understanding. Line 221 "Figure 3" should be "Table". Table 2 is appearing twice and lines 179 and lines 189.

Table 3 comparing the pre-pandemic and during the pandemic is inaccurate for the table heading. What statistical tool was used to determine the p-values?

Authors' attention are urgently called to re-analyze Table 3 and use frequency data rather than percentages because these do not represent the data required to explain the table, especially if the statistical tool used is the Chi-square test of goodness-of-fit, which should be the most appropriate. The various categories do not add up to 100% for each response categories. If these variables were scored then apply accordingly.

Data analysis require a second review. Multinomial logistic regression is not clear for this study noting the objective of the study. If this analysis is relevant, then revise the title and objectives to reflect factors associated with changes in "...dietary and physical activity pre-pandemic and during the pandemic".

Reviewers' comments:

Reviewer's Responses to Questions

**Comments to the Author**

1. Does this manuscript meet PLOS Global Public Health’s publication criteria? Is the manuscript technically sound, and do the data support the conclusions? The manuscript must describe methodologically and ethically rigorous research with conclusions that are appropriately drawn based on the data presented.

Reviewer #1: Yes

Reviewer #2: Yes

2. Has the statistical analysis been performed appropriately and rigorously?

Reviewer #1: Yes

Reviewer #2: No

3. Have the authors made all data underlying the findings in their manuscript fully available (please refer to the Data Availability Statement at the start of the manuscript PDF file)?

Reviewer #1: No

Reviewer #2: No

4. Is the manuscript presented in an intelligible fashion and written in standard English?

Reviewer #1: Yes

Reviewer #2: Yes

5. Review Comments to the Author

Reviewer #1: There are few instances where some words are used interchangeably. There is a need for consistency. For instance, individuals recruited into quantitative study are referred to as respondents. While those recruited into qualitative study are referred to as participants

There are some statements that show no agreement between them. The authors should clarify this.

There is missing information in the methods on how data was collected.

The results section needs a revisit.

The discussion and the conclusions need modification after the major issues in the result section are addressed.

Many typos and grammar errors

The attached comments will help the authors.

Reviewer #2: This study investigated the changes in dietary habits and physical activity of diabetic patients during the COVID-19 pandemic and compared the results to self-reported status at pre-pandemic. This would have been an excellent study if it was appropriately designed and implementation adequately articulated. The concept is excellent but there are serious flaws needing to be corrected to strengthen the manuscript. Below are my comments and suggestions for revision.

These comments are made based on the 1.) The title of the study, 2.) the objectives of the study and therefore, if the authors’ decide to retain these two premise for the review would need to change the title and include modifications to the study objectives.

The abstract to this study needs to be revised to communicate adequately what the study sought to achieve and what was found. The following represent revised suggestion. (See attached review for suggestions).

Introduction

The introduction require revision to strengthen understanding of the context involved. (See attached review for suggestions).

Materials and methods require very careful appraisal of study design, participant enrolment into the study, instrument development and validity, data analysis

The results in the manuscript needs to be organized to facilitate clarity and understanding. Line 221 "Figure 3" should be "Table". Table 2 is appearing twice and lines 179 and lines 189.

Table 3 comparing the pre-pandemic and during the pandemic is inaccurate for the table heading. What statistical tool was used to determine the p-values?

Please check the attached document of detailed comments and suggestions.

6. PLOS authors have the option to publish the peer review history of their article (what does this mean?). If published, this will include your full peer review and any attached files.

**Do you want your identity to be public for this peer review?** For information about this choice, including consent withdrawal, please see our Privacy Policy.

Reviewer #1: No

Reviewer #2: **Yes: **Bola Christie Atulomah

---

## [Editor Report · Decision Letter 1]

2 Jun 2023

Changes in Dietary Habits and Physical Activity among the Diabetic Patients of Bangladesh during COVID-19: A Cross-Sectional Study

PGPH-D-23-00054R1

Dear  Dr.Sharif ,

We are pleased to inform you that your manuscript 'Changes in Dietary Habits and Physical Activity among the Diabetic Patients of Bangladesh during COVID-19: A Cross-Sectional Study' has been provisionally accepted for publication in PLOS Global Public Health.

Best regards,

Nnodimele Onuigbo Atulomah, PhD

Academic Editor

Congratulations for completing the recommended revision of the manuscript submitted.